# VprBP/DCAF1 Triggers Melanomagenic Gene Silencing through Histone H2A Phosphorylation

**DOI:** 10.3390/biomedicines11092552

**Published:** 2023-09-17

**Authors:** Yonghwan Shin, Sungmin Kim, Gangning Liang, Tobias S. Ulmer, Woojin An

**Affiliations:** 1Department of Biochemistry and Molecular Medicine, Norris Comprehensive Cancer Center, University of Southern California, Los Angeles, CA 90033, USA; yonghwas@usc.edu (Y.S.); skim7438@usc.edu (S.K.); 2Department of Urology, Norris Comprehensive Cancer Center, University of Southern California, Los Angeles, CA 90089, USA; gliang@usc.edu; 3Department of Physiology and Neuroscience, Zilkha Neurogenetic Institute, University of Southern California, Los Angeles, CA 90033, USA; tulmer@usc.edu

**Keywords:** VprBP, phosphorylation, melanoma, histone, H2A, kinase

## Abstract

Vpr binding protein (VprBP), also known as DDB1- and CUL4-associated factor1 (DCAF1), is a recently identified atypical kinase and plays an important role in downregulating the transcription of tumor suppressor genes as well as increasing the risk for colon and prostate cancers. Melanoma is the most aggressive form of skin cancer arising from pigment-producing melanocytes and is often associated with the dysregulation of epigenetic factors targeting histones. Here, we demonstrate that VprBP is highly expressed and phosphorylates threonine 120 (T120) on histone H2A to drive the transcriptional inactivation of growth-regulatory genes in melanoma cells. As is the case for its epigenetic function in other types of cancers, VprBP acts to induce a gene silencing program dependent on H2AT120 phosphorylation (H2AT120p). The significance of VprBP-mediated H2AT120p is further underscored by the fact that VprBP knockdown- or VprBP inhibitor-induced lockage of H2AT120p mitigates melanoma tumor growth in xenograft models. Collectively, our results establish VprBP-mediated H2AT120p as a key epigenetic signal for melanomagenesis and suggest the therapeutic potential of targeting VprBP kinase activity for effective melanoma treatment.

## 1. Introduction

Vpr binding protein (VprBP) was initially identified as the cellular target and binding protein of the human immunodeficiency virus type 1 (HIV-1) accessory protein viral protein r (Vpr) and also named DDB1- and CUL4-associated factor1 (DCAF1) [1]. Since its discovery, VprBP has been mainly implicated in the process of regulating cell cycle and proliferation as a substrate recognition component of the DNA damage-binding protein 1 (DDB1)-Cullin 4 (Cul4)-RING finger protein (ROC1) E3 ubiquitin ligase complex [2,3,4,5]. However, our former study revealed an additional function for VprBP as an effector that binds nucleosomes and inactivates transcription in the context of chromatin [6]. More surprisingly, a further investigation of the mechanism behind VprBP-mediated transrepression uncovered the presence of intrinsic kinase activity in VprBP and established T120 of histone H2A as the first physiological substrate [7]. VprBP inactivates transcription dependently of H2A T120 phosphorylation (H2AT120p), because VprBP kinase-dead mutation (K194R) and H2AT120p blocking mutation (T120A) abrogate VprBP transrepression activity in our functional assays [7,8]. In accordance with the idea that VprBP-mediated H2AT120p is an oncogenic signal, data from our gene expression profiling in colon and prostate cancer cell lines clearly indicate that targeting and silencing growth-regulatory genes reflects the primary role of VprBP in cancer cells [7,8]. Moreover, our Western blot and immunohistochemical analyses of cancer cell lines and tumor samples highlighted a direct relationship between elevated expression of VprBP and increased levels of H2AT120p [7,8]. Considering that VprBP kinase activity toward H2AT120 plays a causal role in tumorigenesis, we have also developed a small-molecule inhibitor, named B32B3, capable of targeting the VprBP catalytic domain, attenuating H2AT120p, and blocking tumor growth, even causing some partial tumor regression, in colon and prostate cancer xenograft models [7]. While the above results highlight the role of VprBP-mediated H2AT120p in the development of colon and prostate cancers, the contribution of VprBP to other types of cancer and the potential causative relationship between H2AT120p and cancer development still remain to be determined.

Melanoma is a serious type of skin cancer and most often generated by the malignant transformation of melanin pigment-producing melanocytes in the bottom layer of the skin epidermis [9,10,11,12]. The pathogenesis of melanoma transformation, referred to as melanomagenesis, is a complex process and is a result of stepwise alterations in transcriptional pathways by a variety of mechanisms [13,14,15]. Among these mechanisms are the direct mutations of gene regulators such as BRAF, neurofibromatosis 1 (NF1), and rat sarcoma virus (RAS) as well as changes in their expression levels that trigger transcriptional perturbations and alter downstream signaling pathways [16,17,18,19,20]. Since all genes conferring a high risk of developing melanoma are associated with histone proteins and stored in the nucleus by chromatinization, and since chromatin regulates the transcriptional competency of all cellular genes, it is also likely that chromatin-dependent pathways account for fundamental mechanisms underlying melanomagenesis. This interesting possibility has not been investigated in any great detail, but there is some indirect evidence linking chromatin reorganization to transcriptional misregulation in melanoma cells. For example, histone deacetylases are often dysregulated in the process of melanomagenesis, and their inhibitors can reactivate tumor suppressor genes to reverse melanoma proliferation and viability [21,22,23]. Enhancer of zeste 2 polycomb repressive complex 2 subunit (EZH2), which has histone methyltransferase activity, is highly expressed and installs the H3K27me3 histone mark at tumor suppressor genes, thereby inhibiting their expression to drive the development and metastasis of melanoma [24,25]. Also, adding another pathway through which melanomagenesis is epigenetically regulated, our recent study identified a role for matrix metalloproteinase 9 (MMP-9) in clipping histone H3 N-terminal tails and conferring active expression properties to genes encoding growth-stimulatory factors [26]. The observed role of MMP-9 has potential therapeutic implications since treatment with MMP-9 inhibitor can restore a silenced state of aberrantly activated genes and exert antagonistic effects on melanoma formation [26]. Hence, the continuous identification and characterization of factors capable of triggering the epigenetic perturbation of gene-regulatory networks are crucial for preventing and curing melanoma skin cancer.

In this study, we employed a combination of genome-wide transcriptome profiling, ChIP-qPCR, CRISPR-dCas9 system, and in vivo xenograft models to investigate a possible functional contribution of VprBP toward melanomagenesis. Our data show that VprBP localizes at genes regulating cell growth and establishes their transcriptional incompetence thereby positively influencing melanoma development. VprBP executes its pro-melanomagenic function in a H2AT120p-dependent manner, as selective mutation and pharmacological inhibition eliminating VprBP kinase activity restore the expression of aberrantly silenced growth-regulatory genes in melanoma cells. Using in vivo models, we further demonstrate that VprBP-mediated H2AT120p is directly linked to melanoma tumor growth, providing an unprecedented documentation of VprBP function as a driver of melanomagenesis as well as an underlying mechanism of action both in vitro and in vivo.

## 2. Materials and Methods

### 2.1. Cell Lines, Constructs, and Antibodies

NHEM2 cells were cultured in 1× Melanocyte Growth Medium M3 with 1× SupplementMix, and G361, MeWo, SK-MEL-5, and A375 cells were cultured in Dulbecco’s modified Eagle’s medium containing 10% fetal bovine serum (FBS). For mammalian expression of VprBP, the VprBP cDNA was amplified by PCR and ligated into the lentiviral expression vector pLenti-Hygro (Addgene, Watertown, MA, USA) containing 5′ 3 × FLAG coding sequence. To generate VprBP K194R expression vector, wild-type VprBP cDNA was mutated by using Q5 Site-Directed Mutagenesis Kit (New England Biolabs, Ipswich, MA, USA) after the construction. Antibodies used in this study are as follows: anti-Histone H2AT120p antibody from Active Motif (Carlsbad, CA, USA); anti-Actin and anti-HA antibodies from Proteintech (Rosemont, IL, USA); anti-VprBP and anti-Lamin B1 antibodies from Thermo Fisher Scientific (Waltham, MA, USA); and anti-Histone H2A antibody from Abcam (Cambridge, MA, USA).

### 2.2. Protein and Histone Extraction and Western Blotting

Whole cell lysates were prepared from G361, MeWo, SK-MEL-5, A375, and NHEM2 cells using M-PERTM Mammalian Protein Extraction Reagent (Thermo Scientific, Waltham, MA, USA) according to the manufacturer’s instructions. Total histone proteins were acid-extracted from the cultured cells and Western blot assays with prepared samples were conducted as described previously [26].

### 2.3. RNA Interference

DNA oligonucleotides (5′-CGAGAAACTGAGTCAAATGAA-3′) encoding shRNA specific for VprBP coding region were annealed and ligated into the lentiviral expression vector pLKO.1 (Addgene, Berkeley, CA, USA). Lentiviral particles were generated in 293T cells by transfecting plasmids encoding VSV-G, NL-BH, and the shRNA. Two days after transfection, the soups containing the viruses were collected and used to infect G361, MeWo cells in the presence of polybrene (8 µg/mL). The cell lines were selected for two weeks in the presence of puromycin (2 µg/mL). For rescue experiments of ectopic VprBP expression, VprBP-depleted cells were infected with lentiviruses expressing shRNA-resistant VprBP wild-type or kinase-dead mutant K194R and selected for two weeks in the presence of hygromycin (300 µg/mL).

### 2.4. Immunostaining

Patient tissue slides containing melanoma and adjacent normal samples were obtained from Novus Biologicals (Centennial, CO, USA). Formalin-fixed, paraffin-embedded sections of these tissue samples were exposed to the blocking reagent (50 mM Tris-HCl, pH 7.5, 150 mM NaCl, 0.3% Triton X-100, and 5% normal goat serum) for 30 min at room temperature. The sections were then incubated with H2AT120p and VprBP antibodies overnight at 4 °C. Immunodetection was performed using ABC reagent (Thermo Fisher Scientific, Waltham, MA, USA) according to the manufacturer’s protocol. DAB was utilized for color development, and hematoxylin was used for counterstaining. For immunostaining of G361 and MeWo cells, the cells were cultured in 4-well chamber slides and fixed with 4% paraformaldehyde for 15 min. The cells were then permeabilized with 0.1% Triton X-100 for 15 min and treated with the blocking reagent for 60 min at room temperature. After incubating with H2AT120p antibody overnight and Alexa Fluor-conjugated secondary antibody for 60 min, the cells were then washed with PBS and imaged using a fluorescence microscope (BZ-X; Keyence, San Diego, CA, USA).

### 2.5. RNA-Seq

RNA was extracted from G361 cells using the Qiagen RNeasy kit (Qiagen Inc., Valencia, CA, USA) according to the manufacturer’s instructions. RNA quality was assessed using an Agilent Bioanalyzer with the DNA1000 kit. Strand-specific libraries were generated from the isolated RNA using the KAPA Stranded mRNA-Seq Kit with KAPA mRNA Capture Beads (Kapa Biosystems Inc., Wilmington, MA, USA). The resulting libraries were pooled, denatured, and diluted to 15 pM before clonal clustering onto the sequencing flow cell using the Illumina cBOT Cluster Generation Station and Cluster Kit v3-cBot-HS. The clustered flow cell was sequenced using 1 × 50 SE reads on the Illumina HiSeq according to the manufacturer’s protocol. Base conversion was performed using OLB version 1.9, and the resulting sequences were demultiplexed and converted to Fastq using CASAVA version 1.8 (Illumina, San Diego, CA, USA) [26,27]. The sequenced RNA-seq reads were then aligned to the hg38 GENCODE version 29 using STAR 2.6.1d (National Human Genome Research Institute (NHGRI), Bethesda, MD, USA) [26,27,28]. The aligned reads were quantified at the gene level, and gene counts were normalized using the upper quartile normalization method. Principal component analysis with normalized gene counts was performed, and differentially expressed genes were selected using the Gene Specific Algorithm from Partek Flow software, https://www.partek.com/partek-flow/ (Partek Inc., Chesterfield, MO, USA). A volcano plot was generated using fold change and false discovery rate of genes, with a false discovery rate cutoff of 0.05 and absolute fold change >2 to statistically detect significantly differentially expressed genes. Gene ontology analysis of differentially expressed genes was performed using the Ingenuity Pathway Analysis tool (IPA version 52912811) (Qiagen Inc., Valencia, CA, USA). Heatmaps were generated by calculating the Z score of gene expression levels using the Generalized Minimum Distance R package heatmap.3 function [29].

### 2.6. Functional Enrichment Analysis and Visualization

The significantly upregulated genes (log2FC ≥ 1, padj < 0.05) were subjected to functional annotation through the Gene ontology (GO) and Reactome pathway, utilizing clusterProfiler (ver. 4.6.2) [30] and ReactomePA [31] packages in the R software (version 4.2.3; R Foundation for Statistical Computing, Vienna, Austria). The data were filtered with the Benjamini–Hochberg method, and statistical significance was classified at a *p*-value and q-value threshold of <0.05. To simplify the interpretation of functional enrichment, a network-based enrichment pathway was displayed by Cytoscape (ver. 3.9.1) application EnrichmentMap (ver. 3.3.5) [32,33,34]. The gene sets related to the biologically similar functions were clustered and annotated using Cytoscape plug-in AutoAnnotate (ver. 1.4.0). The list of genes was filtered by an FDR cutoff of <0.05 and *p*-value threshold of 0.05. Subsequently, a post-analysis in the EnrichmentMap application was conducted to explore potential connections between our functional enrichment map and apoptotic signature genes. To perform this analysis, the apoptosis signature was obtained from Human MSigDB v2023. 1. Hs [35] and the overlap with these gene sets was scored using Fisher’s exact test with a significance threshold of *p*-value < 0.05.

### 2.7. RT-qPCR

Total RNA was isolated from G361 and MeWo cells by using the RNeasy Mini kit (Qiagen Inc., Valencia, CA, USA) and converted to first-strand cDNA using the SuperScript III First-Strand System Kit (Thermo Fisher Scientific, Waltham, MA, USA). Real-time RT-qPCR was performed with SYBR Green Real-time PCR Master Mixes (Thermo Fisher Scientific, Waltham, MA, USA) according to the manufacturer’s instructions. The primers used for RT-qPCR are listed in Appendix A. All mRNA values were normalized to GAPDH mRNA levels, and all reactions were run in triplicate.

### 2.8. ChIP-qPCR

ChIP assays with G361 and MeWo cells were performed using the ChIP Assay Kit (Millipore, Burlington, MA, USA) as recently described [8,36]. After reversing protein–DNA crosslinks, immunoprecipitated DNA was purified and analyzed by qPCR using the primers that amplify the promoter (P), transcription start site (TSS), and coding (C) region of INPP5J, ZNF750, and TUSC1 genes. The primers used for qPCR are listed in Appendix A. Specificity of amplification was determined by melting curve analysis, and all samples were run in triplicate.

### 2.9. Cell Viability and Colony Formation Assays

VprBP-depleted/rescued G361 and MeWo cells were seeded into 96-well plates at a density of 2 × 10^3^ cells/well, and cell viability was evaluated over a period of 5 days by MTT assay. To determine the effects of VprBP inhibitor, MTT assays were also conducted after treating G361 and MeWo cells with 1 µM of VprBP inhibitor B32B3 for 5 consecutive days. For colony formation assays, G361 and MeWo cells were seeded at a density of 500 cells/well in a 6-well plate and treated with different concentrations of B32B3 for 72 h, and cells were allowed to form colonies for an additional 2 weeks. The colonies in each well were stained with 0.5% crystal violet and photographed. The colonies in each well were counted using ImageJ 1.45s software (U.S. National Institutes of Health, Bethesda, MD, USA). All assays were run in triplicate, and the results presented are the average of three individual experiments.

### 2.10. CRISPR/dCas9-Based Kinase Assays

The dCas9-VprBP fusion protein expression vector was generated by fusing human VprBP to cDNA encoding the catalytically inactive nuclease codon-optimized S. pyogenes Cas9 (dCas9) in an expression vector using the CAG promoter. The pPlatTET-gRNA2 vector (Addgene 82559) was digested and gel-extracted using a Qiagen gel extraction kit. VprBP wt or VprBP K194R was cloned from complementary DNA (cDNA) using primers. The digested and gel extracted pPlatTET-gRNA2 vector was then ligated with VprBP wt or VprBP K194R according to the manufacturer’s instructions. To construct sgRNAs, INPP5J, ZNF750, and TUSC1 gene datasets were selected from UCSC and used to design Crispr sgRNAs, 58 bp oligos including specific sgRNA sequences, by CHOPCHOP (https://chopchop.rc.fas.harvard.edu, accessed on June 2022). The oligos designed were synthesized for PCR amplification with primers (Appendix A). The amplified fragments were purified and used for a recombination reaction according to the In-Fusion HD Cloning Plus Kit protocol (Takara Bio Inc., San Jose, CA, USA) with the pPlatTET-gRNA2 construct digested. VprBP-depleted G361 cells (cell number) were transfected with plasmids encoding dCas9, dCas9- VprBP wt or dCas9-VprBP K194R by using FuGENE^®^ HD Transfection Reagent (Promega, Madison, WI, USA). Cells were passaged 48 h post-transfection, and 50 μg/mL neomycin was added 3 h after plating. Media were exchanged 2 days post-transfection and cells were passaged every other day starting 4 days after initial replating. Neomycin selection was maintained for a total of 7 days. Western blotting, RT-qPCR, cell viability, and colony formation assays were performed to study kinase, transrepressive, and growth-regulatory activities of dCas9-VprBP fusions in transfected cells.

### 2.11. Mice Xenograft

In vivo experiments were performed using athymic male NCr-nu/nu mice (Charles River, Wilmington, MA, USA), and the mice were maintained under specific pathogen-free conditions. To evaluate the impact of knockdown, four groups, each comprising six mice, were randomly assigned, while six groups, each consisting of six mice, were randomly allocated to investigate the effect of B32B3 treatment (5 mg/kg). G361 melanoma cells were subsequently subcutaneously injected into the right hind leg of each mouse, and tumor growth was monitored by measuring body weights every 3 days for a duration of 24 days. Tumor volumes were estimated by measuring the width (W) and length (L) of the tumor using a digital caliper and calculated based on the following formula: TV = W2L/2. At the end point of experiments, mice were sacrificed by asphyxiation with CO_2_, and G361 melanoma xenografts were excised, photographed, and weighed. To determine the levels of VprBP and H2AT120p, lysates were also prepared from the excised xenografts and analyzed by Western blotting. All animal experiments were performed according to protocols approved by the Institutional Animal Care and Use Committee.

### 2.12. Statistical Analysis

All quantitative data are presented as mean ± standard deviation (SD). Statistical analyses of datasets were performed with two-way ANOVA or Student’s two-tailed *t*-test followed by Bonferroni post hoc test using GraphPad Prism 9 software (GraphPad Software Inc., Boston, MA, USA). A *p* value < 0.05 was considered statistically significant.

## 3. Results

### 3.1. VprBP/DCAF1 Is Overexpressed and Catalyzes H2AT120p in Melanoma Cells

Because VprBP expression is often dysregulated in cancer cells [7,8], we reasoned that aberrant expression of VprBP could also be observed in melanoma cells. To explore this possibility, cell lysates were prepared from four melanoma (G361, MeWo, SK-MEL-5, and A375) and one melanocyte (NHEM2) cell line and analyzed by Western blotting with the VprBP antibody. Our analysis detected VprBP expression at much higher levels in the melanoma cells compared to the melanocyte cells (Figure 1A). In checking whether the observed VprBP overexpression was associated with altered H2AT120p, we observed much higher levels of H2AT120p in chromatin fractions extracted from the melanoma cells. Further, the stable depletion of VprBP with shRNA knockdown almost completely eliminated H2AT120p, and these changes could be rescued by the ectopic expression of VprBP wild-type but not VprBP K194R kinase-dead mutant in G361 and MeWo melanoma cells, as assessed by immunostaining and Western blot analyses (Figure 1B,C and Appendix A). As another approach to check a possible link between VprBP and H2AT120p in melanoma cells, we carried out immunostaining with melanoma and adjacent normal tissue samples. As shown in Figure 1D, our immunostaining demonstrated that the VprBP protein level was significantly elevated in melanoma patient tissue samples, and its expression was highly correlated with H2AT120p. Adding support for this observation, our analysis of VprBP expression levels in five stages of melanoma detected VprBP overexpression in all five stages of melanoma with only minor variations (Appendix A).

We previously screened a library of 5000 compounds and identified B32B3 as a selective inhibitor targeting VprBP catalytic domain and thus blocking VprBP kinase activity [7]. Given the demonstrated reliance of H2AT120p on VprBP in melanoma cells, it was reasonable to expect that B32B3 treatment would recapitulate the effects of VprBP knockdown. Toward this end, we treated G361 and MeWo cells with increasing concentrations of B32B3, and evaluated changes in H2AT120p by Western blot. The range of B32B3 concentrations used in these initial assays is based on our recent studies to determine its IC50 values using colon and prostate cancer cell lines [7,8]. When the melanoma cells were exposed to six different concentrations (0, 0.03, 0.1, 0.3, 1, and 3 µM) of B32B3, B32B3 was able to potently block H2AT120p with a half-maximal inhibitory concentration (IC50) of 0.1 µM (Figure 1E and Appendix A). Consistent with these results, our immunostaining of G361 and MeWo cells after B32B3 treatment at the IC50 concentration detected a significant reduction in H2AT120p levels (Figure 1F and Appendix A). Together, these initial observations link VprBP overexpression to H2AT120p in melanoma cells and rationalize further studies on their possible contributions to melanomagenesis.

### 3.2. VprBP/DCAF1 Knockdown and Inhibition Suppress Melanoma Cell Growth

The data presented above confirmed that VprBP is overexpressed and is responsible for H2AT120p event in melanoma cells; however, the potential significance of VprBP-mediated H2AT120p with respect to melanoma cell growth remains unclear. In an effort to address this question, we monitored changes in cell growth rates in response to VprBP depletion daily over a period of five days using MTT assays. As summarized in Figure 2A and Appendix A, our assays revealed an apparent decrease in cell growth rate after the stable knockdown of VprBP in G361 and MeWo melanoma cells. To examine whether VprBP capacity to mediate H2AT120p is necessary for the observed effects, we also conducted rescue experiments. The expression of VprBP wild-type restored the original growth rate of VprBP-depleted melanoma cells. On the other hand, VprBP kinase dead mutant failed to recover the growth capacity of VprBP-depleted G361 and MeWo cells (Figure 2A and Appendix A), underscoring the notion that VprBP-mediated H2AT120p is critical for VprBP function in promoting melanoma cell growth. For the purpose of gaining support for the MTT assay results, we also evaluated the impact of VprBP knockdown on melanoma cell growth using colony formation assays. The results from these experiments indicate that VprBP depletion adversely affects the potential of melanoma cells to grow into a colony when evaluated after 14 days of culture (Figure 2B and Appendix A). Since VprBP-mediated H2AT120p displays a sharp reduction in G361 and MeWo melanoma cells treated with VprBP inhibitor B32B3 (Figure 2C and Appendix A), we wondered whether the growth of melanoma cells is also suppressed by such B32B3 treatments. In initially exploring this possibility, we discovered that exposing G361 and MeWo melanoma cells to B32B3 caused a marked impediment to the growth ability of the cells (Figure 2C and Appendix A). In performing colonogenic assays with G361 and MeWo cells, we also observed the colony-forming capacity of the cells to reduce by up to 40% after B32B3 treatment (Figure 2D and Appendix A). These results underscore the direct action of VprBP on melanomagenesis and support the concept that VprBP exerts its melanomagenic function via H2AT120p.

### 3.3. VprBP/DCAF1-Mediated H2AT120p Inactivates Growth-Regulatory Genes in Melanoma Cells

In order to examine whether the VprBP-mediated H2AT120p described above plays any role in regulating melanomagenic transcription program, genome-wide RNA sequencing (RNA-seq) analysis was performed with total RNA isolated from control and VprBP-depleted G361 cells. In the principal component analysis (PCA) of RNA-seq data, samples for each group were found to be markedly separated from each other, but close clustering of replicates from groups indicated minimal variability in the quality of analyzed replicates (Figure 3A). Using a 2-fold cutoff, our transcriptome profiling revealed a total of 1941 genes differentially expressed upon stable knockdown of VprBP in melanoma cells (Figure 3B,C). Among those genes, 674 genes were downregulated and 1267 genes upregulated in response to VprBP knockdown (Figure 3C and Appendix A). Consistent with our previous publications implicating VprBP-mediated H2AT120p in oncogenic gene silencing, gene ontology analysis of 1267 upregulated targets also identified cell growth and proliferation as the most activated biological pathways in VprBP-depleted melanoma cells (Figure 3D). Our Gene Ontology (GO) and Reactome pathway enrichment analyses of the genes that were upregulated upon VprBP knockdown revealed a significant enrichment of metabolic process genes (Appendix A). The functional enrichment analysis using a list of significantly enriched GO terms also links VprBP to metabolic processes (Appendix A). Given the well-established importance of metabolic processes in melanoma development and progression [37], these findings strongly implicate VprBP in the pathogenesis of melanoma. The role for VprBP in melanomagenesis was further supported by the fact that our analysis of the leading-edge subset in the gene set detected 20 genes encoding negative regulators of melanoma growth and proliferation (Figure 3E). To validate our RNA-seq data, we conducted a reverse transcription quantitative PCR (RT-qPCR) analysis of eight target genes that encode factors acting as tumor suppressors in several types of cancers including melanoma (Figure 4A and Appendix A). Our analysis with total RNA from G361 and MeWo cells demonstrated the up- and downregulation of the selected targets after VprBP knockdown and rescue expression (Figure 4A and Appendix A), respectively. Importantly, if VprBP K194R kinase-dead mutant was expressed in VprBP-depleted cells, target genes were still expressed at high levels, underscoring the importance of VprBP-mediated H2AT120p for target gene inactivation in melanoma cells. Similar RT-qPCR assays after treatment with VprBP inhibitor B32B3 also detected the active state of target genes in G361 and MeWo cells (Figure 4C and Appendix A), resulting in the incapability of VprBP to induce H2AT120p and thus target gene depotentiation.

To check whether the observed function of VprBP in suppressing target gene transcription reflects its stable recruitment, we next investigated its occupancy at INPP5J, ZNF750, and TUSC1 genes by chromatin immunoprecipitation (ChIP) assays. Crosslinked chromatin was isolated from control and VprBP-depleted G361 and MeWo cells, and the precipitated DNA was amplified by quantitative real-time PCR (qPCR) using primer sets specific for promoters, transcription start sites, and coding regions of INPP5J, ZNF750, and TUSC1 genes. Consistent with our previous observation [7,8], VprBP ChIP signals were much more enriched at the promoter region than at transcription start site and coding region in mock-depleted control melanoma cells (Figure 4B and Appendix A). This result is supportive of the idea that VprBP targets the process of initiating transcription for its repressive action. VprBP distribution patterns across the target genes were perfectly correlated with H2AT120p enrichment patterns, suggesting a major role for VprBP in mediating H2AT120p at target genes. Indeed, the VprBP occupancy of the target genes was significantly reduced after VprBP knockdown, and such changes also diminished the levels of H2AT120p. It was also apparent in our parallel ChIP-qPCR assays that the ectopic expression of VprBP wild-type, but not VprBP K194R kinase-dead mutant, in VprBP-depleted cells restored H2AT120p to levels quantitatively similar to those observed with mock-depleted control cells (Figure 4B and Appendix A). These observations were further corroborated by additional ChIP- and RT-qPCR experiments in which exposure of G361 and MeWo melanoma cells to VprBP inhibitor B32B3 almost completely crippled H2AT120p and triggered target gene reactivation (Figure 4D and Appendix A). Taken together, these data present a persuasive argument that VprBP-mediated H2AT120p is a necessary step to downregulate growth-regulatory genes and to drive melanomagenesis.

### 3.4. Artificial Tethering of VprBP/DCAF1 to Target Genes Drives H2AT120p-Induced Transrepression

Through RNA-seq and ChIP/RT-qPCR studies described above, we identified a group of growth-regulatory genes inactivated by VprBP and enriched for H2AT120p in melanoma cells. These results support the conjecture that H2AT120p mainly contributes to VprBP-induced transcriptional suppression leading to melanomagenesis. However, these studies do not exclude the possibility that VprBP generates the inactive state of target genes through some other mechanisms. If VprBP exerts its suppressive effects mainly by catalyzing H2AT120p at target genes, we can predict that artificially tethering VprBP to target genes is sufficient to re-establish an inactive state of transcription in VprBP-depleted melanoma cells. In exploring this possibility, we chose to use CRISPR/dCas9-based system for directing VprBP-mediated H2AT120p to target genes [38,39,40,41]. For this cellular manipulation of H2AT120p, we constructed a series of pPlatTET-gRNA2 all-in-one vectors expressing dCas9-VprBP wild-type (wt) or kinase dead mutant (K194R) and single-guide RNAs (sgRNAs) recognizing the promoters or coding regions of INPP5J, ZNF750, and TUSC1 genes (Figure 5A). We then tested the impact of these dCas9-VprBP fusions on the transcription of INPP5J, ZNF750, and TUSC1 genes in G361 melanoma cells. As summarized in Appendix A, the individual expression of promoter-binding sgRNA 1 and 2 together with dCas9-VprBP wild-type in VprBP-depleted G361 cells led to a detectable inactivation of target gene transcription, whereas coding region-binding sgRNA 3 or 4 generated no obvious changes in transcription. Also, by directing dCas9-VprBP to the promoter region with sgRNA 1 and 2 pair, we were able to generate a more pronounced repression of INPP5J, ZNF750, and TUSC1 genes (Appendix A). Under identical assay conditions, sgRNA 3 and 4 pair minimally altered the levels of target gene transcription, thus implying that transcription initiation is the step mainly regulated by VprBP-mediated H2AT120p (Appendix A). Consistent with expectations from these results, targeting dCas9-VprBP to both promoter and proximal coding regions by sgRNA 1, 2, 3 and 4 together established target gene silencing at levels comparable with those observed with sgRNA 1 and 2 pair (Appendix A). In additional experiments, B32B3 treatment significantly impaired the suppressive activity of dCas9-VprBP wild-type at INPP5J, ZNF750, and TUSC1 genes and dCas9-VprBP K194R kinase-dead mutant failed to exert any effects on transcription (Figure 5B), clearly indicating the requirement of VprBP kinase activity for target gene silencing in melanoma cells.

To gain support for the transcription results, we next investigated dCas9-VprBP localization and its impact on H2AT120p in the promoter and coding regions of target genes by ChIP-qPCR analysis. The expression of dCas9-VprBP wild-type or K194R kinase-dead mutant with sgRNA 1 and 2 pair generated a specific accumulation of the dCas9 fusion proteins in the promoter regions of the INPP5J, ZNF750, and TUSC1 genes (Figure 5C). When sgRNA 1 and 2 were replaced by coding region-binding sgRNA 3 and 4, the dCas9-VprBP fusions were mainly localized in the proximal coding regions. In monitoring the extent of H2AT120p, we found that H2AT120p levels were increased in the promoter regions in VprBP-depleted G361 cells expressing dCas9-VprBP wild-type together with sgRNA 1 and 2 pair (Figure 5C). Parallel analysis on the proximal coding regions repeatedly demonstrated an efficient accumulation of H2AT120p in the cells transfected with dCas9-VprBP wild-type and sgRNA 3 and 4 (Appendix A). As with another evidence supporting the precise targeting of dCas9-VprBP fusions, H2AT120p was established at both promoter and coding regions after simultaneously transfecting dCas9-VprBP wild-type and all four sgRNAs into the cells (Appendix A). However, in agreement with transcription data, all the sgRNAs failed to increase H2AT120p levels at the target genes in dCas9-VprBP K194R-expressing cells as well as B32B3-treated cells.

Given that the INPP5J, ZNF750, and TUSC1 genes encode the components of the growth control system, we subsequently tested whether delivering dCas9-VprBP with sgRNA 1 and 2 to their promoter regions can lead to changes in cell growth rate. In our MTT and colony formation assays, a marked increase in cell growth rates and colony numbers was evident, when dCas9-VprBP wild-type was co-expressed with promoter-binding sgRNA 1 and 2 in VprBP-depleted G361 cells (Figure 5D,E). Conversely, the co-expression of dCas9-VprBP K194R kinase-dead mutant with sgRNA 1 and 2 failed to trigger a similar augmentation in the growth and colony formation potential of VprBP-depleted G361 cells. Since B32B3 treatment also almost completely abolished the ability of dCas9-VprBP wild-type to repotentiate the growth activity of sgRNA 1 and 2-transfected G361 cells (Figure 5D,E), these results strongly suggest that VprBP can function to stimulate the growth of melanoma cells in a kinase-activity-dependent manner. Together with ChIP-qPCR data above, these results also discount the possibility that VprBP drives the melanomagenic gene silencing program in an H2AT120p-independent manner, and reinforce the conclusion that VprBP can accurately establish H2AT120p-induced gene silencing and growth-stimulatory effects if stably recruited to the target genes.

### 3.5. VprBP/DCAF1 Promotes Melanoma Tumorigenesis in Its Kinase-Activity-Dependent Manner

Based on our demonstration of VprBP being overexpressed and stimulating melanoma cell growth, an important question is whether VprBP knockdown or inhibition exhibits anti-melanomagenic efficacy through blocking VprBP kinase activity toward H2AT120 and thus decreasing VprBP transrepression potential. As a way to address this question, we decided to use xenograft mouse models derived from the G361 melanoma cell line. Accordingly, we subcutaneously injected 1 × 10^7^ mock-depleted control or VprBP-depleted G361 cells into the right hind leg of nude mice, and monitored their growth at 3-day intervals for a period of 24 days. From this first set of experiments, we found that VprBP depletion significantly inhibited the growth of G361 melanoma xenografts when compared with mock-depleted control G361 xenografts (Figure 6A–C). Moreover, the expression of VprBP wild-type, but not VprBP K194R mutant, in VprBP-depleted G361 xenografts resulted in a full recovery of the original xenograft growth rate (Figure 6A–C), indicative a major role for VprBP-mediated H2AT120p in stimulating melanoma development and progression. To support the knockdown data, we also tested whether treatment with increasing concentrations (0.5, 1, 2.5, 5 and 10 mg/kg) of VprBP inhibitor B32B3 would influence the growth of G361 xenografts. As can be seen in Figure 7A–C, the proliferative capacity of G361 melanoma xenografts was reduced by an average of 70% following the administration of 2.5 mg/kg B32B3 every 3 days for 24 days. At two higher doses of B32B3 (5 and 10 mg/kg), a greater impairment of melanoma growth compared with a dose of 2.5 mg/kg B32B3 was not observed (Figure 7A–C). Also, that VprBP knockdown and B32B3 treatment were well tolerated without any significant changes in body weight (Appendix A) argues strongly that their inhibitory effects are generated by specifically targeting G361 melanoma cells in mice.

Another key question arising from the above-noted xenograft growth data is to what extent VprBP knockdown and inhibition affect H2AT120p in G361 melanoma xenograft models. In an attempt to address this question, we sacrificed the mice, harvested melanoma xenograft tumors, and prepared xenograft lysates. Our Western blot analysis detected high levels of H2AT120p in lysates collected from control G361 melanoma xenografts (Figure 6E and Figure 7E). However, H2AT120p was almost completely disappeared upon shRNA-mediated knockdown and B32B3-induced inhibition of VprBP (Figure 6E and Figure 7E), again suggesting that VprBP stimulates the growth of G361 xenograft in a manner dependent on its kinase activity toward H2AT120. Considering the importance of H2AT120p in VprBP-driven transcriptional silencing of growth-regulatory genes, we also examined whether VprBP depletion and inhibition-induced blocking of VprBP kinase activity also affected target gene expression in G361 xenografts. We found that VprBP knockdown generated, albeit to a somewhat varying extent, an active state of target gene expression at the level of transcription in G361 melanoma xenografts (Figure 6D). Consistent with the melanoma-promoting effects of VprBP-mediated H2AT120p, treating G361 melanoma xenograft with VprBP inhibitor B32B3 at doses of 2.5 mg/kg or higher over a period of 24 days also markedly increased target gene mRNA levels (Figure 6D and Figure 7D). B32B3 thus recapitulates the anti-melanomagenic effects of VprBP knockdown and represents a potent molecular tool to negate VprBP-induced melanoma development and proliferation.

## 4. Discussion

VprBP-driven tumorigenesis has been studied with much greater attention to its action as an adaptor of the DDB1-Cul4-ROC1 E3 ubiquitin ligase complex [3,5]. However, our recent studies have uncovered the intrinsic kinase activity residing in VprBP and established H2AT120 as its first cellular target [7]. Our molecular and genome-wide characterization of VprBP using colon and prostate cancer cells indicate that VprBP overexpression and consequent H2AT120p downregulate a group of growth-regulatory genes, suggesting its direct contribution to epigenetic gene silencing and cancer pathogenesis [7,8]. Despite the role of VprBP in colon and prostate cancers established in these earlier studies, its possible involvement in increasing the risk of other types of cancer is still not clear. In this report, we demonstrate that the H2AT120p activity of VprBP is directly linked to altered transcription program during the process of melanoma development.

In our initial effort to understand the possible role of VprBP-mediated H2AT120p in melanomagenesis, we compared the gene expression profiles of VprBP-depleted melanoma cells with those of mock-depleted cells. There were significant changes in the expression of growth-regulatory genes after VprBP knockdown, and the altered genes were especially enriched with functions that contribute to cell growth control, generally acting to inhibit cell proliferation and cancer development. Among the most significantly altered genes, we selected INPP5J, ZNF750, and TUSC1 for further study because their gene products are known to reduce the viability and malignant proliferation of melanoma cells. In probing their promoter regions, transcription start sites, and proximal coding regions by ChIP, we detected higher levels of VprBP occupancy and thus H2AT120p at the promoter region compared to the coding region and TSS. With the use of VprBP inhibitor B32B3, we also confirmed a major role of VprBP-mediated H2AT20p in inactivating growth-regulatory genes as well as stimulating melanoma tumor growth in vivo. These findings are in good agreement with those of our previous reports, demonstrating that VprBP mainly localizes to target gene promoters and generates locus specific H2AT120p to modulate their transcriptional competence in colon and prostate cancer cells [7,8]. Our current work also provides strong support for the epigenetic action of VprBP dependent of H2AT120p in driving gene-silencing programs necessary for melanomagenesis and represents a significant extension of its biological contribution as a histone kinase (Appendix A). In fact, the findings described in this study provide the first demonstration that histone modification directly contributes to melanoma tumorigenesis through the mis-regulation of gene transcription (Appendix A). Nonetheless, many interesting questions remain to be answered. For instance, it is unknown whether other histone-modifying activities, in addition to VprBP, are involved in the formation of constitutively repressed states of anti-melanomagenic genes. Because some histone modifications such as acetylation activate transcription, opposite to the trans-repressive effects of H2AT120p, future work examining melanomagenic transcription program should also focus on the competitive action of factors depositing active histone marks against VprBP. Related to this, our recent investigation revealed an additional role for VprBP in phosphorylating EZH2 in colon cancer cells and established another epigenetic process underlying a VprBP-induced oncogenic gene silencing program [42]. While the potential involvement of VprBP-mediated EZH2 phosphorylation in melanomagenesis was not explored in the current study, the stable depletion and inhibition of VprBP in melanoma cells expressing low levels of EZH2 still generated a substantial increase in melanomagenic gene transcription and cell growth rates, lending support to the notion that the pro-melanomagenic activity of VprBP is mainly through H2AT120p at target genes. Thus, it is tempting to speculate that VprBP on the one hand acts through H2AT120p and, on the other hand, regulates through EZH2 phosphorylation for its tumorigenic activity in a cell-type-specific manner.

Another important finding in this study was generated through our CRISPR experiments demonstrating that dCAS9-VprBP along with sgRNA sets is able to initiate a long-term repression at three representative target genes: INPP5J, ZNF750, and TUSC1. Although there have been reports correlating H2AT120p to distinct transcriptional status, the use of the CRISPR-dCAS9 system allowed us to provide the first direct connection between H2AT120p and target gene inactivation. Also supportive of the idea that VprBP modulates transcription at the level of initiation, dCAS9-VprBP was highly efficient in suppressing the transcription of INPP5J, ZNF750, and TUSC1 genes when guided to their promoter regions by sgRNAs in VprBP-depleted melanoma cells. However, targeting dCAS9-VprBP to proximal coding regions of INPP5J, ZNF750, and TUSC1 genes was much less efficient in suppressing their transcription. Perhaps more important is the observation that there is a strong correlation between the level of dCas9-VprBP-mediated H2AT120p and the level of dCas9-VprBP-induced transcriptional inactivation and that dCas9-VprBP kinase dead mutant is unable to switch growth-regulatory genes from an active to an inactive transcriptional state in melanoma cells. This observation suggests that the presence and function of VprBP at promoter regions is critical for the stable inactivation of growth-regulatory genes in melanoma cells. These findings also raise the question of how VprBP-mediated T120p of nucleosomal H2A acts as a negative regulator for gene transcription. There are two potential explanations for the trans-repressive effect of H2AT120p observed in melanoma cells. H2A is the only core histone containing a long carboxyl-terminal tail, and deletion of the tail domain significantly inhibits the interaction between H2A-H2B dimer and H3-H4 tetramer [43]. Therefore, it is likely that H2AT120p at its C-terminal extension alters the interaction between H2A-H2B dimer and H3-H4 tetramer or between DNA and H2A-H2B dimer in the nucleosome. The crystallographic structure of the nucleosome also indicates that the C-terminal domain of H2A is present in close proximity to the site where the DNA enters and exits the nucleosome [44,45,46]. Thus, one possibility implies that VprBP-mediated H2AT120p influences nucleosome structure by increasing the affinity of H2A-H2B dimer for the H3-H4 tetramer and inducing a conformational change in the nucleosome. Related to the second possibility, evidence from recent studies demonstrated that H2AT120p can serve as a binding platform for the stable tethering and function of gene-regulatory factors at target chromatin regions [47,48,49,50,51]. Considering such observations, another possibility is that specific effector proteins are recruited to growth-regulatory genes by recognizing H2AT120p present on promoter nucleosomes. This stable localization of effectors across the promoter regions of VprBP target genes establishes the inactive state of transcription, especially impeding transcription initiation, in melanoma cells. Structural and molecular investigations of H2AT120p nucleosome will provide information on the nature of H2AT120p-induced intra/inter-nucleosomal reorganization and how VprBP-mediated H2AT120p creates a transcriptionally unfavorable chromatin environment in melanoma cells. In this regard, targeting therapeutic intervention to VprBP activity toward H2AT120p-induced chromatin compaction and factor recruitment could provide an effective treatment strategy for melanoma.

## 5. Conclusions

In the present study, we demonstrate that VprBP is overexpressed and plays a key role in transcriptionally inactivating a group of growth-regulatory genes in melanoma cells. VprBP executes these functions in a manner dependent on H2AT120p, as inhibitor treatment and kinase-dead mutation almost completely abolishes the gene silencing potential of VprBP. For further support for such an H2AT120p-dependent function, artificial tethering of VprBP wild-type, but not kinase-dead mutant, to target genes is sufficient for achieving an inactive transcriptional state. These results thus reveal a hitherto-unknown role of VprBP in the development of melanoma, as well as the molecular mechanism involved in the observed action of VprBP by linking H2AT120p to an oncogenic gene silencing program. Also, our observation that B32B3 treatment significantly inhibits melanoma growth is supportive of the idea that the development of more potent VprBP inhibitors has potential implications in terms of melanoma-targeted therapy.

## Figures and Tables

**Figure 1 biomedicines-11-02552-f001:**
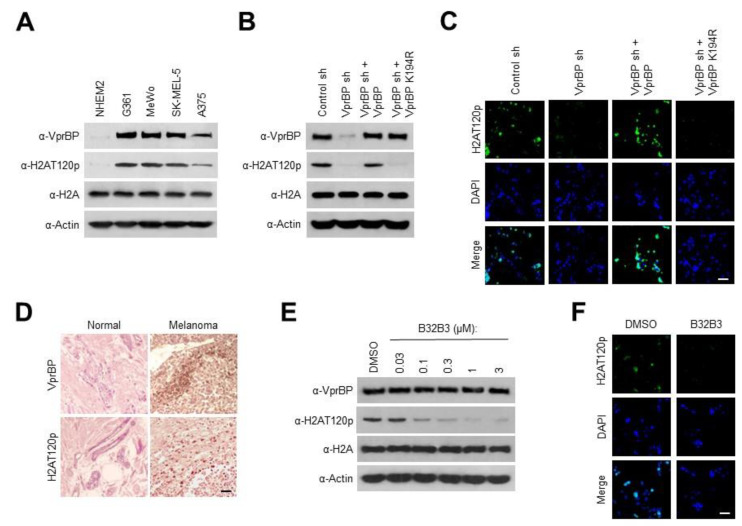
VprBP is overexpressed and mediates H2AT120p in melanoma cells. (**A**) Whole cell lysates and chromatin fractions were prepared from melanoma (G361, MeWo, SK-MEL-5, and A375) and melanocyte (NHEM2) cells and analyzed by Western blotting with VprBP, H2AT120p, and H2A antibodies. Actin served as a control for equal protein loading. A representative blot of three independent experiments is displayed. (**B**) G361 cells were transfected with nontargeting control (control) or VprBP shRNA, and whole cell lysates and chromatin fractions were analyzed by Western blotting with the indicated antibodies. VprBP-depleted cells were also complemented by shRNA-resistant VprBP wild-type or kinase-dead mutant K194R to check their rescue effects. Actin was used as a loading control. (**C**) VprBP-depleted G361 cells were transfected with VprBP wild-type or K194R and immunostained with H2AT120p antibody. Representative images of three independent experiments are shown. Bar, 10 µM. (**D**) Human normal skin and melanoma tissues were subjected to immunohistochemistry with antibodies against VprBP and H2AT120p. High power magnifications are shown for representative immunostaining samples. Bar, 50 µm. (**E**) G361 cells were grown in the presence of the indicated concentrations of VprBP inhibitor B32B3 for 72 h. Western blot analysis of cell lysates and chromatin prepared from the G361 cells with VprBP, H2AT120p, H2A, and Actin antibodies. Shown are the representative results of three independent experiments. (**F**) G361 cells were immunostained with H2AT120p antibody to monitor relative changes in H2AT120p levels after treating with B32B3 as in (**E**). Images are representative of three independent experiments.

**Figure 2 biomedicines-11-02552-f002:**
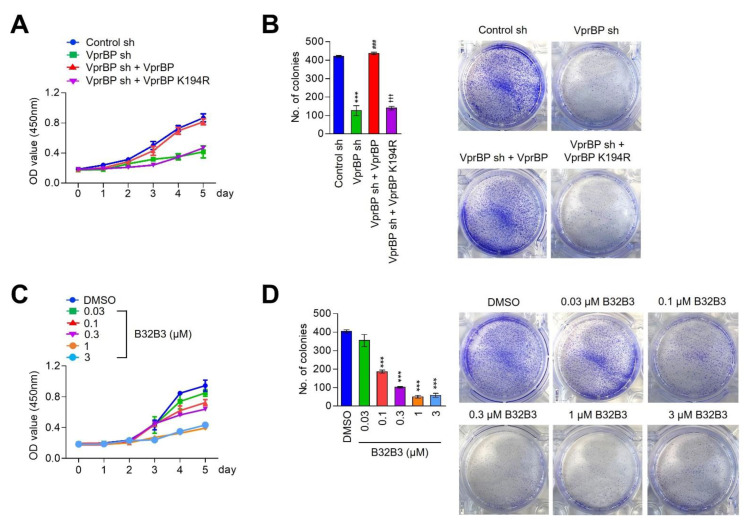
VprBP downregulation restricts melanomagenesis. (**A**) VprBP-depleted G361 cells were complemented with VprBP wild-type or kinase-dead mutant, and their proliferation was assessed after 5 days of culture by MTT assay. Results represent the mean ± SD of three experiments performed in triplicate. (**B**) The indicated G361 cells were allowed to form colonies for 2 weeks in 6-well plates, stained with Crystal violet, and counted. Data represent the mean ± SD of three independent experiments in triplicate well; *p* values were calculated using paired *t*-tests. *** *p* < 0.001 versus control sh; ^###^
*p* < 0.001 versus VprBP sh; and ^†††^
*p* < 0.001 versus control sh. (**C**) G361 cells were treated with VprBP inhibitor B32B3 for 5 days, and their viability was measured by MTT assay. Data represent the mean ± SD of three independent experiments performed in triplicate. (**D**) Colony formation assays were conducted as in B, but after treating with VprBP inhibitor B32B3 for 5 days. Data are representative of three independent experiments performed in triplicate and represent the mean ± SD; *p* values were calculated using paired *t*-tests. *** *p* < 0.001 versus DMSO.

**Figure 3 biomedicines-11-02552-f003:**
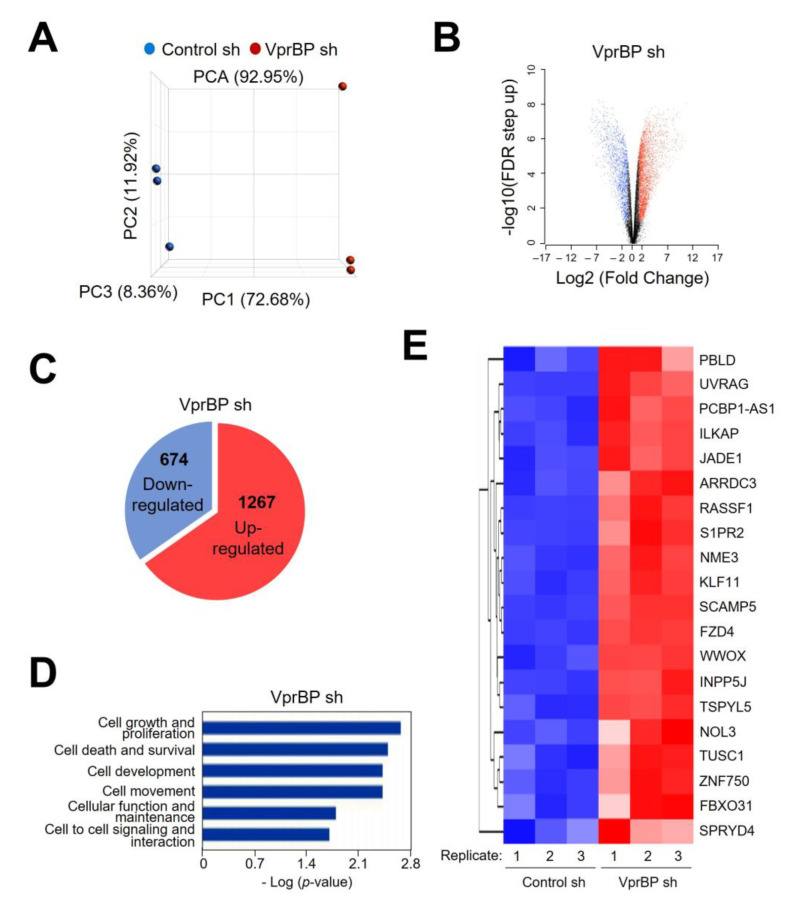
VprBP impairs the expression of growth-controlling genes. (**A**) Principal component analysis (PCA) results of RNA-seq datasets generated in G361 melanoma cells. VprBP knockdown (VprBP sh) group is shown in red, and control (control sh) group is shown in blue. Three replicates were generated per group. (**B**) A volcano plot of RNA-seq datasets is shown. −log10 (FDR step up) is shown on the Y-axis, and fold change in gene expression between VprBP knockdown and control groups is shown on the X-axis. Genes modulated after VprBP depletion are colored in blue (downregulated) and red (upregulated). (**C**) Venn diagram showing genes that are upregulated or downregulated (>2 fold; FDR < 0.05) in VprBP-depleted G361 cells compared to mock-depleted control cells. (**D**) Gene ontology analysis of the activated genes after knockdown of VprBP using Ingenuity Pathway Analysis (IPA version 52912811) tool developed by Qiagen. (**E**) Heatmap of 20 genes most activated upon VprBP depletion. Normalized gene expression levels (Z-scores) are plotted. High and low expression are shown in red and blue, respectively.

**Figure 4 biomedicines-11-02552-f004:**
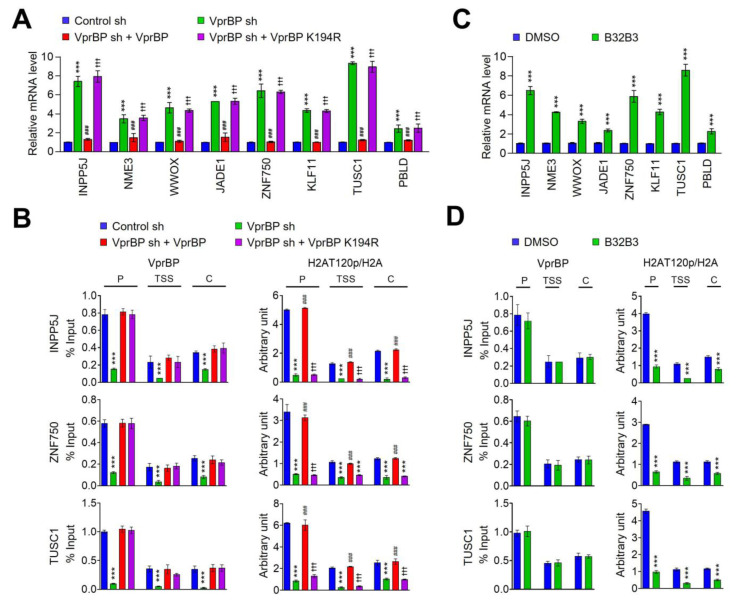
VprBP target genes are enriched for H2AT120p. (**A**) RNA samples were prepared from mock-depleted control, VprBP-depleted, or wild-type/mutant VprBP-transfected VprBP-depleted G361 cells and analyzed by RT–qPCR using primers listed in Appendix A. All transcription levels were normalized to that of GAPDH. Data are expressed as mean ± SD (N = 3); *p* values were calculated using paired *t*-tests. *** *p* < 0.001 versus control sh; ^###^
*p* < 0.001 versus VprBP sh; and ^†††^
*p* < 0.001 versus control sh. (**B**) ChIP assays were performed in control, VprBP-depleted, and VprBP-complemented VprBP-depleted G361 cells with VprBP, H2AT120p, and H2A antibodies as indicated. All ChIP DNAs were analyzed by real-time PCR with primer pairs amplifying the promoters, transcription start sites, and coding regions of the INPP5J (upper panel), ZNF750 (middle panel), and TUSC1 (lower panel) genes. Primers used are listed in Appendix A. Error bars denote the mean ± SD obtained from triplicate real-time PCRs. All transcription levels were normalized to those of GAPDH. Data were expressed as mean ± SD (N = 3); *** *p* < 0.001 versus control sh; ^###^
*p* < 0.001 versus VprBP sh; and ^†††^
*p* < 0.001 versus control sh. (**C**) G361 cells were treated with VprBP inhibitor B32B3 for 72 h, and VprBP target gene expression was analyzed by RT-qPCR as in A. Data were expressed as mean ± SD (N = 3); *p* values were calculated using paired *t*-tests. *** *p* < 0.001 versus DMSO. (**D**) ChIP assays were performed as in B but using B32B3-treated G361 cells. Data were expressed as mean ± SD (N = 3); *** *p* < 0.001 versus DMSO.

**Figure 5 biomedicines-11-02552-f005:**
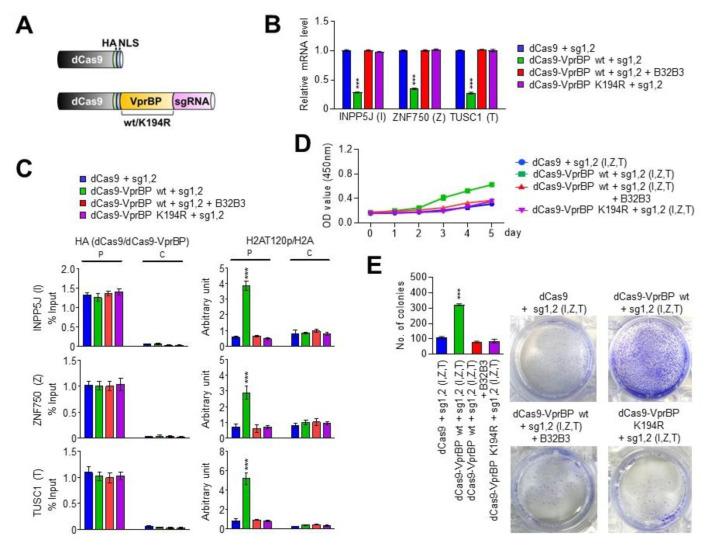
dCas9-VprBP fusion specifically inactivates target gene transcription. (**A**) Schematic diagram of the CAG-based constructs driving the expression of dCas9 or dCas9—VprBP wt/K194R kinase dead mutant and sgRNAs targeting INPP5J, ZNF750, and TUSC1 genes. (**B**) VprBP-depleted G361 cells were transfected with the indicated dCas9 and sgRNA expression constructs for 48hr, and total RNA was isolated and analyzed by qRT-PCR using primers specific for the INPP5J, ZNF750, and TUSC1 genes. Data represent the mean ± S.D. (N = 3); *** *p* < 0.001 versus control. (**C**) VprBP-depleted G361 cells were transfected with dCas9 and sgRNA expression constructs as in (**B**), and the levels of H2AT120p at the promoter and coding regions of the INPP5J, ZNF750, and TUSC1 genes were assessed by ChIP-qPCR. Data represent the mean ± S.D. (N = 3); *** *p* < 0.001 versus control. (**D**) VprBP was guided to target genes by using CRISPR/dCas9 system as in C and changes in cell growth were assessed by MTT assays over a period of 5 days. The results represent the mean  ±  SD of three experiments performed in triplicate. (**E**) Colony formation assays were carried out with G361 cells after selectively downregulating INPP5J, ZNF750, and TUSC1 genes by CRISPR/dCas9 system as in B. Data represent the mean  ±  SD of three independent experiments in triplicate wells; *** *p*  <  0.001.

**Figure 6 biomedicines-11-02552-f006:**
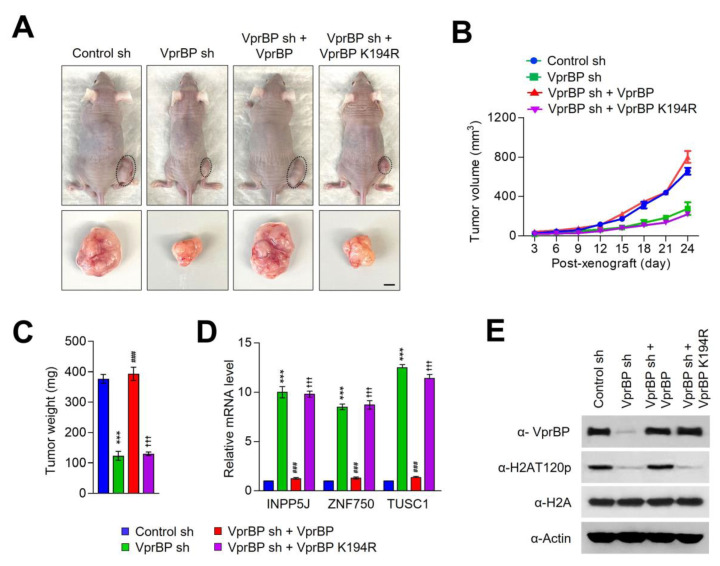
VprBP knockdown reduces melanoma tumor growth in vivo. (**A**) Mock-depleted control, VprBP-depleted, or wild-type/mutant VprBP-transfected VprBP-depleted G361 cells were injected into the right side of mouse skin. Mice were sacrificed 24 days after G361 cell injection, and melanoma xenografts were surgically excised and photographed (lower panel, scale—1 cm). (**B**) Melanoma xenograft tumor volume was measured every 3 days after injecting G361 cells into mice as in (**A**). (**C**) G361 melanoma xenografts were excised as in (**A**), and their weights were measured and expressed in milligrams. Data are presented as the mean ± S.D (N = 6); *** *p* < 0.001 versus control sh; ^###^
*p* < 0.001 versus VprBP sh; and ^†††^
*p* < 0.001 versus control sh. (**D**) Relative mRNA levels of INPP5J, ZNF750, and TUSC1 genes in G361 melanoma xenografts obtained 24 days post-injection were determined by RT-qPCR. Data represent the mean ± S.D. (N = 3); *** *p* < 0.001 versus control sh; ^###^
*p* < 0.001 versus VprBP sh; and ^†††^
*p* < 0.001 versus control sh. (**E**) The excised G361 xenografts were analyzed by Western blotting with the indicated antibodies.

**Figure 7 biomedicines-11-02552-f007:**
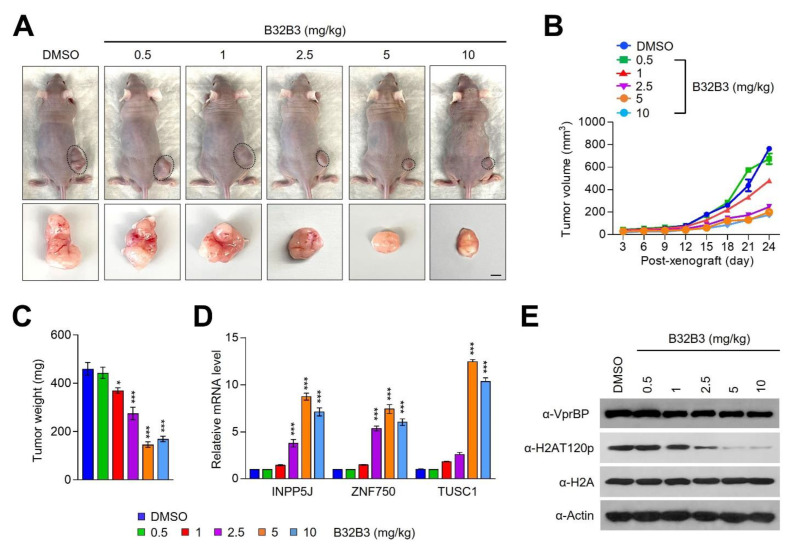
VprBP inhibition reduces melanoma tumor growth in vivo. (**A**) Melanoma xenografts were established as in Figure 6 and treated with DMSO or VprBP inhibitor B32B3 for 24 days. Mice were sacrificed at the end of 24-day B32B3 treatment, and melanoma xenografts were surgically excised and photographed (lower panel, scale: 1 cm). (**B**) The volume of melanoma xenograft tumors was measured every 3 days over a 24-day B32B3 treatment period. (**C**) After treating with B32B3 for 24 days, G361 melanoma xenografts were excised from mice, and tumor weight was measured. Data represent the mean ± S.D. (N = 6); * *p* < 0.05, *** *p* < 0.001 versus DMSO. (**D**) After 24 days of B32B3 treatment, G361 melanoma xenografts were excised as in C and relative mRNA levels of INPP5J, ZNF750, and TUSC1 genes were measured by RT-qPCR. Data represent the mean ± S.D. (N = 3); *** *p* < 0.001 versus DMSO. (**E**) Western blot analysis of the excised melanoma xenografts using the antibodies indicated on the left.

## Data Availability

The gene expression array data were deposited in the NCBI Gene Expression Omnibus (GEO) database under the GEO accession number GSE230573. The data are also available in Appendix A.

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
