# Peer review of "VprBP/DCAF1 Triggers Melanomagenic Gene Silencing through Histone H2A Phosphorylation"

_biomedicines, 2023, doi:10.3390/biomedicines11092552_

Round 1
Reviewer 1 Report
1. Avoid using first-person writing throughout the manuscript.
2. Abstract: Provides full names of all abbreviations at the first time.
3. Keywords: These should be key terms but did not appear in the manuscript title.
4. Figure 1D: A scale bar needs to be added. The color and background look slightly different between the photos and at least two of them are out of focus.
5. Figures 4B and D: Significant difference between means is missing and should be added.
6. Figure 5C: Significant difference between means is missing and should be added.
7. Figure 6E: The quality is problematic in the first and third bands of alpha-actin.
8. Figure 7E: The quality of alpha-VprBP is not good.
9. How many replicates were conducted for each treatment?
10. The authors should organize a conclusive figure to present their hy[othesis in the possible mechanism to enhance the overall quality.
Author Response
We would like to thank you and the Reviewers for the constructive suggestions and comments on our manuscript entitled “VprBP/DCAF1 triggers melanomagenic gene silencing through histone H2A phosphorylation” We are sending you herewith our revised manuscript and response to the Reviewers’ comments. We have addressed all the points raised by the Reviewers and revised the manuscript according to the Reviewers’ suggestions. In view of importance of this work for the biomedical community, we sincerely hope that you will recommend this work for prompt publication in the Biomedicines.
Below, we enumerate our detailed response to the comments made by the Reviewers
Reviewer 1
Point 1: “Avoid using first-person writing throughout the manuscript.”
Response 1: We acknowledge the reviewer's comment regarding the avoidance of first-person writing throughout the manuscript. We have taken the reviewer's comment seriously and conducted a thorough review of the entire manuscript to identify any instances of first-person writing. After this review, we are pleased to confirm that we could not identify any occurrences of first-person writing in the manuscript.
Point 2: “Abstract: Provides full names of all abbreviations at the first time.”
Response 2: We have implemented the required revisions by incorporating the full names of all abbreviations as per reviewer's suggestion. This ensures that readers obtain a comprehensive understanding of our work right from the beginning.
Point 3: “Keywords: These should be key terms but did not appear in the manuscript title.”
Response 3: While it is accurate that three out of the five keywords are integrated into the title, we would like to elucidate our rationale for including all keywords within the abstract. Given the intricacy of the subject matter, compressing all keywords into a single sentence of the title posed challenges. Consequently, we have opted to present all keywords within the abstract, ensuring a comprehensive portrayal of our study.
Point 4: “Figure 1D: A scale bar needs to be added. The color and background look slightly different between the photos and at least two of them are out of focus.”
Response 4: We have addressed all the concerns the reviewer raised regarding Figure 1D; Scale bars have been added to the figure, scale bar sizes have been stated in the figure legends, and all photo colors have been updated to ensure consistency and accuracy.
Point 5: “Figures 4B and D: Significant difference between means is missing and should be added.”
Response 5: We have included the missing information regarding significant differences between means in response to reviewer’s suggestion.
Point 6: “Figure 5C: Significant difference between means is missing and should be added.”
Response 6: As the reviewer suggested, the missing information pertaining to significant differences between means has been incorporated into the revised manuscript.
Point 7: “Figure 6E: The quality is problematic in the first and third bands of alpha-actin.”
Response 7: We have made the necessary adjustments to improve the quality of the alpha-actin bands in the revised manuscript.
Point 8: “Figure 7E: The quality of alpha-VprBP is not good.”
Response 8: In our revision, we have incorporated the necessary adjustments into the figure to enhance the quality and accuracy.
Point 9: “How many replicates were conducted for each treatment?”
Response 9: The number of replicates we conducted for each treatment is clearly stated in the section 2.11 "Mice Xenograft" of the Materials and Methods. Just to reiterate, we employed six mice within each experimental group for all xenograft experiments
Point 10: “The authors should organize a conclusive figure to present their hypothesis in the possible mechanism to enhance the overall quality.”
Response 10: We have included a succinct summary of the VprBP-induced melanomagenic gene silencing in Supplementary Figure S11. This addition serves to comprehensively illustrate our hypothesis and underlying mechanisms for a more cohesive presentation of our work.
Reviewer 2 Report
The authors wrote an interesting paper on VprBP (DCAF1), a recently identified atypical kinase that downregulates tumor suppressor gene transcription and increases cancer risk. The authors demonstrated that it is highly expressed in melanoma cells and phosphorylates threonine 120 (T120) on histone H2A to cause transcriptional inactivation of growth regulatory genes. The entire manuscript is very well organized, including all sections. The figures are well presented and have high resolution to see all the details.Minor comments: Abstract: Explain the abbreviations: VprBD and DCAF1 Introduction: It is well organized. Just please explain the next abbreviations: VrBD immediately at the beginning of the first sentence not later; HIV, DDB1-Cul4-ROC1 E3, NF-1, EZH2, etc. Methods: What does this mean?: "...the mice were maintained under specific pathogen conditions." Please explain.
Author Response
We would like to thank you and the Reviewers for the constructive suggestions and comments on our manuscript entitled “VprBP/DCAF1 triggers melanomagenic gene silencing through histone H2A phosphorylation” We are sending you herewith our revised manuscript and response to the Reviewers’ comments. We have addressed all the points raised by the Reviewers and revised the manuscript according to the Reviewers’ suggestions. In view of importance of this work for the biomedical community, we sincerely hope that you will recommend this work for prompt publication in the Biomedicines.
Below, we enumerate our detailed response to the comments made by the Reviewers
Reviewer 2
“The authors wrote an interesting paper on VprBP (DCAF1), a recently identified atypical kinase that downregulates tumor suppressor gene transcription and increases cancer risk. The authors demonstrated that it is highly expressed in melanoma cells and phosphorylates threonine 120 (T120) on histone H2A to cause transcriptional inactivation of growth regulatory genes. The entire manuscript is very well organized, including all sections. The figures are well presented and have high resolution to see all the details.”
Response: We are pleased to hear that the Reviewer has a very positive view of the manuscript.
Minor comments:
Point 1: “Abstract: Explain the abbreviations: VprBP and DCAF1”
Response 1: The abbreviations "VprBP" and "DCAF1" have now been explained in the abstract of our manuscript, as per reviewer’s recommendation. Specifically, "VprBP" denotes [Vpr Binding Protein], while "DCAF1" refers to [DDB1 and CUL4 Associated Factor 1].
Also see our response to the Point 2 made by the Reviewer 1
Point 2: “Introduction: It is well organized. Just please explain the next abbreviations: VrBD immediately at the beginning of the first sentence not later; HIV, DDB1-Cul4-ROC1 E3, NF-1, EZH2, etc.”
Response 2: In response to this comment, we have explained all the abbreviations when they first show up in the revised manuscript.
Point 3: “Methods: What does this mean?: "...the mice were maintained under specific pathogen conditions." Please explain”
Response 3: Actually it is written as “specific pathogen free conditions”, not “pathogen conditions”, in the Methods section of our manuscript. This refers to the fact that mice were housed under specific pathogen-free (SPF) conditions, which have become the standard environment for maintaining mouse colonies. Mice under SPF conditions are subject to regular screening and protection measures against a range of natural mouse pathogens. This approach minimizes the risk of infections and enhances the reliability and reproducibility of results.
Reviewer 3 Report
This is a very complete manuscript reporting on the role of VprBP histone kinase in melanoma. The authors show that VprBP is highly expressed in melanoma cells, it is active in phosphorylation of histone H2A and evokes silencing of growth regulatory genes. Then they show that both VprBP knockdown and the use of the VprBP inhibitor B32B3 reduces melanoma growth in xenograft models.
I would like to point out several points that need clarification as well as some typos.
1.- The title of section 1 should be Introduction.
2.- Line 7 of Introduction: “our recent study” would be better described as “our former study”, since it is not so recent (2013, i.e.10-year old).
3. The expression “stored in the nucleus by chromatinization” needs a brief explanation or rephrasing.
4.- The expression “Enhancer of zeste 2 polycomb repressive complex 2 subunit (EZH2) histone methyltransferase is highly…” (in the second paragraph of page 2) should be shanged to “Enhancer of zeste 2 polycomb repressive complex 2 subunit (EZH2), which has histone methyltransferase activity, is highly…”
5.- Near the end of section 2.3 add a brief explanation to the shRNA-resistant VprBP: why is it shRNA resistant?
6.- The numbering of supplementary Figures S1 and S2 in the text and in the supplement are not coincident. Figure S1 in the text, corresponds to Figure S2 in the supplement.
7.- Actually, Figure S2 would be better shown as an additional panel of Figure 1 in the main manuscript, perhaps in a form more common and familiar to many readers than the violin plot. Alternatively, the meaning of the violin plot should be explained briefly.
8.- Concerning supplementary material, I strongly suggest some reordering and clarification: Check that all the numbering of supplementary figures and tables coincide in the main manuscript and in the suupplementary files. For instance, Figures S1 and S2 do not coincide. The file "Combined supplementary file-Shin et al." would be much easier to use if each Figure had its own legend in the same page, rather than all the legends together after Figure S11.
9.- Figure 1 legend, parts A, B and E: I understand that what is analyzed is chromatin fractions prepared from whole cell lysates (several times in this legend). However, the expression used in the three parts “whole cell lysates and chromatin fractions” seems to indicate that both lysates and fractions were analyzed.
10. Figure 1 legend, part D: change “50 µM” to “50 µm” (micrometers not micromolar).
11.- Figure 1 legend, part F: it contains a circular reference as it is stated that treatment with B32B3 was done as in F.
12.- The second paragraph of page 14 contains a wrong line break near the beginning.
13.- In Conclusions, colon cancer is mentioned twice, but melanoma is not mentioned.
14.- Institutional Review Board Statement: it is said “Not applicable”, but animal experimentation should be authorized by the relevant body.
Author Response
Reviewer 3
This is a very complete manuscript reporting on the role of VprBP histone kinase in melanoma. The authors show that VprBP is highly expressed in melanoma cells, it is active in phosphorylation of histone H2A and evokes silencing of growth regulatory genes. Then they show that both VprBP knockdown and the use of the VprBP inhibitor B32B3 reduces melanoma growth in xenograft models.
Response: Thanks for your positive comment on our manuscript. We greatly appreciate your time and effort in reviewing our manuscript.
Point 1: The title of section 1 should be Introduction.
Response 1: We have changed the title of section 1 to 'Introduction' as requested by the Reviewer.
Point 2: Line 7 of Introduction: “our recent study” would be better described as “our former study”, since it is not so recent (2013, i.e.10-year old).
Response 2: The Reviewer is right – “our former study” is more appropriate wording. We have changed this accordingly in the revised manuscript.
Point 3: The expression “stored in the nucleus by chromatinization” needs a brief explanation or rephrasing.
Response 3: We accept the Reviewer’s point, and have changed the sentence to “Since all genes conferring a high risk of developing melanoma are associated with histone proteins and stored in the nucleus as a highly compact structure by chromatinization”
Point 4: The expression “Enhancer of zeste 2 polycomb repressive complex 2 subunit (EZH2) histone methyltransferase is highly…” (in the second paragraph of page 2) should be shanged to “Enhancer of zeste 2 polycomb repressive complex 2 subunit (EZH2), which has histone methyltransferase activity, is highly…”
Response 4: We have changed the sentence as the Reviewer suggested in the revised manuscript.
Point 5: Near the end of section 2.3 add a brief explanation to the shRNA-resistant VprBP: why is it shRNA resistant?
Response 5: We have included near the end of section 2.3 a brief explanation on why shRNA-resistant VprBP was utilized in our study.
Point 6: The numbering of supplementary Figures S1 and S2 in the text and in the supplement are not coincident. Figure S1 in the text, corresponds to Figure S2 in the supplement.
Response 6: We apologize for the inconsistency. The related sections have now been corrected in the revised version of the manuscript.
Point 7: Actually, Figure S2 would be better shown as an additional panel of Figure 1 in the main manuscript, perhaps in a form more common and familiar to many readers than the violin plot. Alternatively, the meaning of the violin plot should be explained briefly.
Response 7: We thank the reviewer for these thoughtful comments. However, after thorough discussion, we have decided to keep Figure S2 in the supplementary files as originally planned. Since Figure S2 doesn’t significantly influence our main conclusions, we believe that presenting it as a supplementary material is appropriate for our manuscript's overall structure. In response to Reviewer’s concern about the clarity of the violin plot, we also have ensured that the meaning of the violin plot is properly explained in Section 3.1.
Point 8: Concerning supplementary material, I strongly suggest some reordering and clarification: Check that all the numbering of supplementary figures and tables coincide in the main manuscript and in the suupplementary files. For instance, Figures S1 and S2 do not coincide. The file "Combined supplementary file-Shin et al." would be much easier to use if each Figure had its own legend in the same page, rather than all the legends together after Figure S11.
Response 8: Thanks for this valuable comment concerning the organization of our combined supplementary file. As the Reviewer suggested, (1) we have checked and ensured that supplementary figure/table numbers coincide in both the main manuscript and the supplementary files, and (2) we have restructured the supplementary material file by keeping figures with own legends on the same page.
Point 9: Figure 1 legend, parts A, B and E: I understand that what is analyzed is chromatin fractions prepared from whole cell lysates (several times in this legend). However, the expression used in the three parts “whole cell lysates and chromatin fractions” seems to indicate that both lysates and fractions were analyzed.
Response 9: We appreciate reviewer’s attention to our description in these legends. Actually we tried to explain that we used whole cell lysates for VprBP/Actin blotting and chromatin fractions for H2A/H2AT120p blotting. We apologize for the insufficient way our description of Western blot analyses was written and we have revised the sentences in all related sections to clarify it.
Point 10: Figure 1 legend, part D: change “50 µM” to “50 µm” (micrometers not micromolar).
Response 10: Thank you for catching this error; we have made the correction as suggested.
Point 11: Figure 1 legend, part F: it contains a circular reference as it is stated that treatment with B32B3 was done as in F.
Response 11: We have made the necessary correction to address this issue. We appreciate your keen attention to detail, and your feedback has contributed to the clarity and accuracy of our manuscript.
Point 12: The second paragraph of page 14 contains a wrong line break near the beginning.
Response 12: We have corrected this error.
Point 13: In Conclusions, colon cancer is mentioned twice, but melanoma is not mentioned.
Response 13: Thank you for bringing this to our attention, and we apologize for the oversight. We now have made all necessary corrections.
Point 14: Institutional Review Board Statement: it is said “Not applicable”, but animal experimentation should be authorized by the relevant body.
Response 14: Thank you for your attention to the Institutional Review Board Statement in our manuscript. We have obtained an approval from the institution for our animal experimentation and have submitted the approval letter to the editor. The appropriate information has now been added to the Institutional Review Board Statement to reflect this authorization.
Reviewer 4 Report
In this study, the employed a combination of genome-wide transcriptome profiling, ChIP-qPCR, CRISPR-dCas9 system, and in vivo xenograft models to investigate a possible functional contribution of VprBP toward melanomagenesis. Rich data are actually provided. Data interpretations and discussions are nicely done. I am positive to recommend publication of this work in Biomedicines. It is scientifically nice work and I have some suggestions on rather general points. Please see below.
1) It is better to provide one initial figure to explain outline of this work. Without such guidance figure, this manuscript may not be so friendly to non-specialist readers.
2) Although this manuscript reports lots of important facts, conclusion descriptions are not so sufficient. Not limited to summary descriptions, descriptions on research impacts and perspectives of futures induced by this research have to be aggressively added.
Author Response
Reviewer 4
In this study, the employed a combination of genome-wide transcriptome profiling, ChIP-qPCR, CRISPR-dCas9 system, and in vivo xenograft models to investigate a possible functional contribution of VprBP toward melanomagenesis. Rich data are actually provided. Data interpretations and discussions are nicely done. I am positive to recommend publication of this work in Biomedicines. It is scientifically nice work and I have some suggestions on rather general points. Please see below.
Response: We are pleased to find that the reviewer now has a very positive view of our manuscript
Point 1: It is better to provide one initial figure to explain outline of this work. Without such guidance figure, this manuscript may not be so friendly to non-specialist readers.
Response 1: We have included a succinct summary of the VprBP-induced melanomagenic gene silencing in Supplementary Figure S11. This addition serves to comprehensively illustrate our hypothesis and underlying mechanisms for a more cohesive presentation of our work.
Point 2: Although this manuscript reports lots of important facts, conclusion descriptions are not so sufficient. Not limited to summary descriptions, descriptions on research impacts and perspectives of futures induced by this research have to be aggressively added.
Response 2: Thanks for the Reviewer’s constructive feedback regarding the conclusion section of our manuscript. We have taken this suggestion into consideration and have added the following sentences to the conclusion section to emphasize the possible impacts and future perspectives of our study; “Also our observation that B32B3 treatment significantly inhibits melanoma growth is supportive of the idea that the development of more potent VprBP inhibitors has potential implications in terms of melanoma targeted therapy.”
Round 2
Reviewer 1 Report
It has been revised accordingly.
Author Response
Thank you for confirming our revision.
